# The Social Vulnerability Index, Mortality and Disability in Mexican Middle-Aged and Older Adults

**DOI:** 10.3390/geriatrics6010024

**Published:** 2021-03-08

**Authors:** Natalia Sánchez-Garrido, Sara G. Aguilar-Navarro, José Alberto Ávila-Funes, Olga Theou, Melissa Andrew, Mario Ulises Pérez-Zepeda

**Affiliations:** 1Instituto Nacional de Ciencias Médicas y Nutrición “Salvador Zubirán”, Mexico City 14080, Mexico; sgan30@hotmail.com (S.G.A.-N.); avilafunes@live.com.mx (J.A.Á.-F.); 2Bordeaux Population Health Research Center, INSERM-University of Bordeaux, UMR 1219, F-33000 Bordeaux, France; 3School of Physiotherapy, Dalhousie University, Halifax, NS B3H 4R2, Canada; olga.theou@dal.ca; 4Geriatric Medicine, Department of Medicine, Dalhousie University, Halifax, NS B3H 4R2, Canada; Melissa.Andrew@nshealth.ca (M.A.); mperez@inger.gob.mx (M.U.P.-Z.); 5Instituto Nacional de Geriatría, Mexico City 10200, Mexico

**Keywords:** disability, frailty, geriatric epidemiology, social determinants of health, social vulnerability

## Abstract

The social vulnerability index (SVI) independently predicts mortality and others adverse outcomes across different populations. There is no evidence that the SVI can predict adverse outcomes in individuals living in countries with high social vulnerability such as Latin America. The aim of this study was to analyze the association of the SVI with mortality and disability in Mexican middle-aged and older adults. This is a longitudinal study with a follow-up of 47 months, the Mexican Health and Aging Study, including people over the age of 40 years. A SVI was calculated using 42 items stratified in three categories low (<0.36), medium (0.36–0.47), and high (>0.47) vulnerability. We examined the association of SVI with three-year mortality and incident disability. Cox and logistic regression models were fitted to test these associations. We included 14,217 participants (58.4% women) with a mean age of 63.9 years (±SD 10.1). The mean SVI was of 0.42 (±SD 0.12). Mortality rate at three years was 6% (n = 809) and incident disability was 13.2% (n = 1367). SVI was independently associated with mortality, with a HR of 1.4 (95% CI 1.1–1.8, *p* < 0.001) for the highest category of the SVI compared to the lowest. Regarding disability, the OR was 1.3 (95% CI 1.1–1.5, *p* = 0.026) when comparing the highest and the lowest levels of the SVI. The SVI was independently associated with mortality and disability. Our findings support previous evidence on the SVI and builds on how this association persists even in those individuals with underlying contextual social vulnerability.

## 1. Introduction

As the demographic transition takes place worldwide, novel factors associated with aging health contribute to adverse outcomes [1]. In particular, socio-economic determinants play a crucial role in older adults’ health [2,3]. Older adults have a greater frequency of chronic health conditions and higher mortality rates, as well as other adverse outcomes, and socio-economic determinants are part of the multicausality path that impacts older adults’ health [4]. From a syndemic angle, an adverse health outcome could potentially be caused by the synergic action of biologic and social factor [5].

A number of different aspects have been included in the study of social determinants framework [6]. These factors impact older adults’ health [7,8,9,10,11,12]. Moreover, socio-economic determinants mediate the relationship of frailty with chronic diseases in low- and middle-income countries [13].

An integrative approach to socio-economic determinants in older adults is depicted in the social vulnerability index (SVI). This index gathers a number of social factors (socio-economic status, social capital, isolation, mastery, a sense of control, etc.) into one score ranging from 0 to 1 with a higher score meaning higher social vulnerability, in a similar fashion to that of the frailty index (FI) [14,15]. In various populations worldwide, the SVI predicts mortality and other adverse outcomes such as cognitive decline and disability [16,17,18,19,20].

The link between social vulnerability and frailty has been established previously [14], and previous work by our group has shown a variable relationship between frailty and socio-economic determinants among different age groups, including middle-aged and older adults [21]. Furthermore, social vulnerability has been related to an increased mortality risk even in less frail individuals [18,19,20,21,22]. Inequalities may affect the relationship of SVI with health outcomes. Therefore, the aim of this study is to describe the association of the SVI with a short follow-up mortality and newly established disability in Mexican middle-aged and older adults.

## 2. Materials and Methods

### 2.1. Study Population

This is a secondary analysis of the 2012 and 2015 waves of the Mexican Health and Aging Study [23]. The main purpose of the Mexican Health and Aging Study is to analyze the health dynamics of Mexican older adults. It includes a population-based cohort of community-dwelling Mexican adults aged 50 years or older and their spouses regardless of age (*n* = 15,186). Data was collected in-person at the participant’s home by trained interviewers. Baseline assessments were done in 2001 and there were four additional waves (2003, 2012, 2015, and 2018) [24]. In 2012, a refreshment of the cohort was performed in order to increase the sample that is expected to be followed up until 2021. Each assessment contains information from different domains, including a thorough questionnaire on social determinants and health-related variables that allow the construction of the SVI and FI.

In order to take advantage of the Mexican Health and Aging Study design, all participants in the 2012 wave were included in this study; survivors of the baseline wave in 2012 and the cohort sample refreshment (*n* = 18,465). Information for 2742 and 1275 participants were collected through next-of-kin (spouse, child, person familiar to the surroundings of the deceased in the las months) interviews and proxy interviews, respectively (http://mhasweb.org/StudyDescription.aspx (accessed on 7 March 2021)). These participants as well as those younger than the age of 40 (*N* = 119) or with incomplete data (*n* = 112) were excluded. The final sample included in this study was 14,217. Among them, 924 individuals were lost to follow-up; therefore, 13,293 adults comprised the sample for the survival and disability analysis. (Figure 1)

### 2.2. Exposure Variable, Outcome Variables, and Covariables

Variables included in the SVI were selected according to previously published indexes [14,18,20,21] and included measures from various social domains such as communication, marital status, social support, locus of control, satisfaction with life, social engagement activities and economic status. Forty-two variables (42) were screened, and none had ≥5% of missing values. Each variable was coded into a 0 to 1 scale using two or more categories. For example, self-rated financial status was coded as zero if the response was ‘excellent’, 0.25 if it was ‘very good’, 0.5 if it was ‘good’, 0.75 if it was ‘regular’, and 1 if it was ‘bad’. Once every variable was coded, we counted the number of variables with missing values for each participant and excluded those participants with ≥20% of SVI variables missing (*n* = 48). Finally, to obtain the index, the sum score of all coded variables was divided by the total number of non-missing SVI variables. The higher the score, and the closest to one, the higher the social vulnerability of the individual. For purposes of interpretation, SVI was categorized in three groups (according to tertiles): <0.36, 0.36–0.47, and >0.47.

To construct the frailty index, we followed standard procedures: plotted the frequency of potential deficits with age, examined the deficit prevalence, and excluded variables missing more than 5% of the cases [25]. From the variables available in 2012, 67 were screened and we ended up with a 60-item FI. For the coding and scoring of each variable we followed similar steps with the SVI (see above). Deficits included in the FI, their prevalence and coding are shown in Appendix A. For interpretation purposes, the FI was divided into four categories: <0.1, 0.1–0.2, 0.21–0.3, and >0.3.

Tobacco use was divided into ‘never’, ‘former user’, and ‘current user’. Alcohol drinking was categorized according to the Canadian Centre on Substance Abuse definition. High-risk alcohol drinking was defined as having ≥3 drinks per occasion or ≥15 a week for men, and as having ≥2 drinks per occasion or ≥10 a week for women [26]. Physical activity was measured with the question: “on average in the last two years, have you exercised or performed strenuous activities at least three times a week?” A positive response identified someone as physically active.

Disability was defined as difficulty in performing any of the following activities of daily living (ADL): dressing, walking in a room, bathing, eating, getting in/out of bed, and toileting. Only individuals without disability in 2012 were included in this analysis (*n* = 10,317) and incident disability was considered present if a participant reported difficulty in any ADL in the 2015 assessment.

### 2.3. Statistical Analysis

Descriptive statistics for each variable were done using mean and standard deviations for continuous variables, and relative and absolute frequencies for categorical variables. Kernel distribution plots were drawn stratified by sex for both the SVI and the FI. Four-way interactions were tested for the effect of SVI (independent variables), sex, age and FI, on mortality and disability (dependent variables). No significant interactions were found, and thus analyses were performed without stratification.

Survival was assessed in the 2015 wave. Reliable information on those who died between 2012 and 2015 was gathered from next-of-kin informants. Time in months from the baseline assessment (2012) to the interview in 2015 or death was used to construct the Kaplan–Meier curves and for the Cox regression analyses. The median of follow-up was 47 months.

To establish the ability of the SVI to predict mortality and disability, Receiver operating curves (ROC) were calculated, and we estimated the area under the curve. Furthermore, Kaplan–Meier curves were drawn for the categorical SVI and FI in relation to survival; log-rank tests were conducted. Cox regression (mortality) and logistic regression (incident disability) models were conducted in three sequential steps: the first one was an unadjusted model, the second was a fully adjusted model (age, sex, FI, physically active, tobacco use, high-risk alcohol drinking) and the third included only the variables that remained after the stepwise regression. To test the assumption of proportionality, log–log curves were drawn for all the models. To assess the impact of lost individuals to follow-up, three additional models were conducted, imputing their values as if they were ‘dead’, ‘alive’, or a random outcome. Results were not significantly different (not shown, data available upon request). A *p*-value of 0.05 was considered significant and all the statistical analyses were done in STATA 14.2 (StataCorp LLC, College Station, TX, USA).

## 3. Results

The baseline sample included 14,217 middle-aged and older adults (*n* = 5959 men and *n* = 8379 women), with a mean age of 63.9 years (±SD 10.1). SVI and FI were higher in women compared to men (Table 1). The distribution of the SVI was normal but the FI had a right skewed distribution (Appendix A).

The overall mortality rate was 5.9% (*n* = 785). According to the bivariate analyses, all variables were significantly associated with mortality (Table 2). The group in the highest SVI tertile had a higher rate of mortality compared to the lowest SVI groups (Appendix A). Finally, the AUC of the ROC for the ability of the SVI to predict mortality was 0.659 (Appendix A).

A total of 1367 individuals, out of the 11,846 who had no disability in 2012, developed difficulty in at least one ADL in 2015. All variables were significantly different between those with and without disability (Table 3). The AUC of the ROC for the ability of the SVI to predict disability was of 0.603 (Appendix A).

In the Cox regression unadjusted model, both the intermediate (2.1 (95% CI 1.6–2.7)) and the highest (4.5 (95% CI 3.4–5.8)) SVI tertiles were significantly associated with higher mortality compared to the lowest tertile. In the adjusted model, SVI remained statistically significant only when comparing the highest with the lowest tertiles (1.7 (95% CI 1.2–2.3)). Tobacco use and high-risk alcohol drinking were not significant predictors of mortality, both in the fully adjusted and in the stepwise models (Table 4). In addition, in this last model SVI score highest tertiles remained significant.

Both SVI tertiles were significantly associated with incident disability (unadjusted and adjusted models). In particular, the OR for the intermediate SVI tertile was 1.2 (95% CI 1.1–1.4, *p* = 0.024), and for the highest SVI tertile was 1.3 (95% CI 1.1–1.5, 0.026) when compared to the lowest tertile (Table 4). In the stepwise model, SVI remained significant, and tobacco use with high-risk alcohol drinking were not significantly associated with incident disability.

## 4. Discussion

Social vulnerability is a complex phenomenon, and items included in the SVI are from different psychosocial domains. This study showed the association of the SVI with mortality and incident disability in Mexican middle-aged and older adults; to our knowledge, this is the first study to address this relationship in Latin America. Previous work on SVI has been done in countries with more homogeneous social factors and low inequity (e.g., Canada, USA, and Europe) [14,18,20]. Mexico is a country with well-known high inequality levels, and even in a population with these characteristics, the SVI was associated with mortality and disability. Also, many of the prior studies on SVI did not address the potential impact of lifestyle factors—such as tobacco use, high-risk alcohol drinking, and physical activity—that have been shown to be associated with adverse health outcomes [14,18,19,20,21]. Though in this study, lifestyle factors need to be considered with caution since they are self-reported. In our work, when adjusting using these factors, the SVI’s associations with adverse outcomes persisted. Moreover, according to our results, three domains (apart from the well-known effects of age and sex) are clearly identified as independent risk factors for mortality and disability: frailty, social vulnerability, and physical activity.

Higher social vulnerability was present in our sample and the SVI distribution identified here for Mexico was similar to what Wallace et al. (2015) found in the Mediterranean sub-group of the SHARE study. Since the interactions with FI levels and sex were not significant, the effect of the SVI on mortality and disability was similar across levels of frailty. This is consistent with previous studies showing, a persistent impact of SVI even in the fittest group [18,19,20,21]. Moreover, as stated by Andrew et al. (2012), countries with adverse environmental and infrastructural conditions may have higher mortality rates and small number of very fit individuals. In fact, our sample had no individuals in the zero-state frailty; the lowest FI score was 0.01 (everyone had at least one SVI deficit).

This study has limitations such as, the sample being of a single country could preclude the generalization of our results and may generalize only in contexts similar to Mexico. Furthermore, 6.5% of the individuals were not included in the analyses for various reasons (incomplete data, lost to follow-up, etc.); additional analyses on those excluded did not show different results. Even though our approach in constructing the SVI was consistent with prior operationalizations, some elements are different from previous reports. However, since aging studies around the world include similar questionnaires, the vast majority of included items are similar to other studies. Other social vulnerability factors not included in the SVI that could be specific to a highly disadvantaged population, such as food insecurity or collective violence, could also impact the social vulnerability of the participants in our study; both phenomena have been shown to impact the health of Mexican older adults [27,28].

There is an ongoing discussion about how to measure frailty; however, the FI is a comprehensive measure that maximizes the number of potential deficits included; many of these FI items can also be found in other instruments. Our choice of cutoffs for both the SVI and the FI were only for informative purposes and were not intended to be strict cutoff points; this is typical for the SVI and FI. Initial analysis was done with 0.1 intervals for the SVI score, but due to low sample size for these categories, we decided to use tertiles.

There is an urgent need to focus resources on the improvement of the social aspects of older adults in countries such as Mexico, and to not focus on only the biological aspects. Proper allocation of resources and further research on this matter could shed light on different interventions that change the future of those fast-aging countries such as Mexico.

## 5. Conclusions

Individual-level social vulnerability as measured by the SVI is independently associated with mortality and incident disability in a country with high social vulnerability, such as Mexico.

## Figures and Tables

**Figure 1 geriatrics-06-00024-f001:**
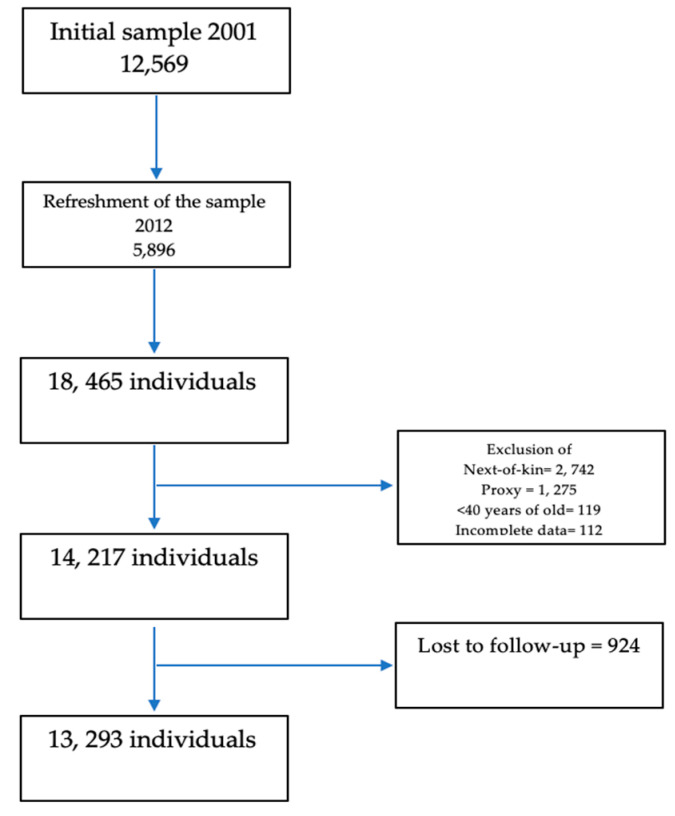
Sample flowchart.

**Table 1 geriatrics-06-00024-t001:** Baseline characteristics of participants stratified by sex.

	Total (*n* = 14,217)	Men (*n* = 5959)	Women (*n* = 8379)
Age, mean (SD)	63.9 (10.1)	65.2 (9.6)	63 (10.2)
Physically active, *n* (%)	5603 (39.4)	2843 (48.1)	2760 (33.2)
Tobacco use, *n* (%)			
Never	8931 (62.8)	2352 (39.8)	6579 (79.2)
Former user	3553 (24.9)	2427 (41)	1126 (13.5)
Current user	1733 (12.1)	1134 (19.1)	599 (7.2)
High-risk alcohol drinking, *n* (%)	1330 (9.3)	1019 (17.2)	311 (3.7)
Social vulnerability index, mean (SD)	0.42 (0.12)	0.39 (0.11)	0.43 (0.12)
Frailty index, mean (SD)	0.23 (0.11)	0.2 (0.1)	0.25 (0.1)

**Table 2 geriatrics-06-00024-t002:** Bivariate analysis of mortality.

	Alive (*n* = 12,508)	Dead (*n* = 785)	*p*-Value *
Age Categories, *n* (%)			
40–49	621 (4.9)	4 (0.5)	<0.001
50–59	3987 (31.8)	57 (7.2)	
60–69	4616 (36.9)	226 (28.8)	
70–79	2488 (19.9)	262 (33.4)	
≥80	796 (6.4)	236 (30.1)	
Sex, *n* (%)			
Women	7390 (59.1)	388 (49.4)	<0.001
Men	5118 (40.9)	397 (50.6)	
Physically active, *n* (%)	5027 (40.2)	177 (22.6)	<0.001
Tobacco use, *n* (%)			
Never	7911 (63.2)	474 (60.3)	0.008
Former user	3076 (24.6)	230 (29.3)	
Current user	1521 (12.1)	81 (10.3)	
High-risk alcohol drinking, *n* (%)	1193 (9.5)	39 (4.9)	<0.001
Social vulnerability index, mean (SD)	0.41 (0.12)	0.48 (0.12)	<0.001
Frailty index, mean (SD)	0.22 (0.1)	0.31 (0.13)	<0.001
Social vulnerability index score tertiles, *n* (%)			
<0.36	4233 (33.8)	126 (16)	<0.001
0.36–0.47	4241 (33.9)	233 (29.7)	
>0.47	4034 (32.2)	426 (54.2)	
Frailty index categories, *n* (%)			
<0.1	1079 (8.6)	21 (2.7)	<0.001
0.11–0.2	4921 (39.3)	168 (21.4)	
0.21–0.3	3525 (28.2)	196 (25)	
>0.3	2983 (23.9)	400 (50.9)	

* *p*-values were obtained from log-rank test; with the exception of the continuous SVI and FI variables in which case a *t*-test was used * *p*-values were obtained from log-rank test; with the exception of the continuous SVI and FI variables in which case a *t*-test was used to estimate *p*-value. * Only those without difficulty in 2012 for any of six activities of daily living (dressing, walking in a room, bathing, eating, going in and out of bed and using the toilet) were included.

**Table 3 geriatrics-06-00024-t003:** Bivariate analysis for incident disability *.

	No Disability (*n* = 8950)	Disability (*n* = 1367)	*p*-Value *
Age Categories, *n* (%)			
40–49	517 (5.8)	34 (2.5)	<0.001
50–59	3310 (36.9)	290 (21.2)	
60–69	3306 (37)	521 (38.1)	
70–79	1485 (16.6)	371 (27.1)	
≥80	332 (3.7)	151 (11.1)	
Sex, *n* (%)			
Women	5069 (56.6)	828 (60.6)	0.006
Men	3881 (43.4)	539 (39.4)	
Physically active, *n* (%)	3924 (43.8)	455 (33.2)	<0.001
Tobacco use, *n* (%)			
Never	5603 (62.6)	902 (66)	0.013
Former user	2178 (24.3)	322 (23.6)	
Current user	1169 (13.1)	143 (10.4)	
High-risk alcohol drinking, *n* (%)	962 (10.7)	116 (8.5)	0.011
Social vulnerability index, mean (SD)	0.2 (0.12)	0.44 (0.12)	<0.001
Frailty index, mean (SD)	0.21 (0.09)	0.28 (0.11)	<0.001
Social vulnerability index score tertiles, *n* (%)			
<0.36	3442 (38.4)	340 (24.8)	<0.001
0.36–0.47	3033 (33.8)	474 (34.6)	
>0.47	2475 (27.6)	553 (40.4)	
Frailty index levels, *n* (%)			
<0.1	681 (7.6)	23 (1.7)	<0.001
0.11–0.2	3924 (43.9)	299 (21.9)	
0.21–0.3	2688 (30)	462 (33.8)	
>0.31	1657 (18.5)	583 (42.6)	

* Only those without difficulty in 2012 for any of six activities of daily living (dressing, walking in a room, bathing, eating, going in and out of bed, and using the toilet) were included.

**Table 4 geriatrics-06-00024-t004:** Multivariate analysis of mortality and incident disability.

	Mortality						Incident Disability					
	Model 1			Model 2			Model 1			Model 2		
	HR	95% CI	*p*-Value	HR	95% CI	*p*-Value	OR	95% CI	*p*-Value	OR	95% CI	*p*-Value
Social vulnerability index tertiles												
<0.36	Reference											
0.36–0.47	1.8	1.4–2.2	<0.001	1.2	0.9–1.5	0.054	1.5	1.3–1.8	<0.001	1.2	1.1–1.4	0.024
>0.47	3.4	2.8–4.1	<0.001	1.4	1.1–1.8	<0.001	2.2	2.6	<0.001	1.3	1.1–1.5	0.026
Age categories												
40–49	Reference											
50–59				1.9	0.7–5.3	0.21				1.3	0.9–1.9	0.17
60–69				5.4	2–14.5	0.001				2.2	1.5–3.1	<0.001
70–79				9.2	3.4–24.9	<0.001				2.9	2–4.3	<0.001
≥80				19.1	7.1–51.7	<0.001				4.8	3.2–7.3	<0.001
Sex												
Women	Reference											
Men				1.7	1.5–2	<0.001				1	0.8–1.1	0.7
Physically active				0.6	0.5–0.7	<0.001				0.8	0.7–0.9	<0.001
Tobacco use												
Never	Reference											
Former user				0.99	0.83–1.18	0.84				0.9	0.8–1.1	0.42
Current user				1.04	0.81–1.3	0.81				0.9	0.7–1.1	0.29
High-risk alcohol drinking				0.7	0.5–1	0.11				1.1	0.9–1.4	0.27
Frailty index levels												
<0.1	Reference											
0.11–0.2				1.2	0.7–1.9	0.05				1.9	1.2–2.9	<0.001
0.21–0.3				1.5	0.9–2.3	0.092				3.7	2.4–5.7	<0.001
>0.31				2.7	1.7–4.2	<0.001				7.2	4.7–11.1	<0.001

HR = hazard ratios, CI = confidence intervals. Model 1: unadjusted. Model 2: adjusted for physical activity, tobacco use and alcohol.

## Data Availability

Data available in a publicly accessible repository http://mhasweb.org/StudyDescription.aspx (accessed on 7 March 2021).

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
