# Peer review of "The Social Vulnerability Index, Mortality and Disability in Mexican Middle-Aged and Older Adults"

_geriatrics, 2021, doi:10.3390/geriatrics6010024_

Round 1
Reviewer 1 Report
This is a longitudinal study investigating the association of the Social vulnerability Index (SVI) with mortality and disability in Mexican middle-aged and older adults. Results from this study suggest that SVI is associated with mortality and disability. Specific comments are below:
Major comments:
It should be clear from the beginning what is your exposure variable (i.e. SVI), what is your outcome variables (i.e. mortality and ADL). As far as I understand, the Frailty Index is “just” a variable that you adjust for. It is given much space in the paper particularly in the introduction – therefore I though in the beginning that it was your outcome variable (i.e. disability). This basic information should be clear to the reader already from the abstract.
Also, it is a bit difficult to follow what you have done. You have plenty of analyses. Perhaps you could exclude some of them e.g. the Kaplan Meier curves and the log rank test (it is enough to have the Hazard ratios).
You must also highlight that it is a longitudinal study, though with a short follow-up of three years (median = 47 months).
Particularly the abstract should be rewritten. Here you should elaborate the following:
- More information about the SVI: 1) A description (you have the description in lines 80-83) 2) more info about the score (i.e. a score from 0-1, higher score meaning higher social vulnerability). The mean SVI of 0.42 is not informative in itself.
- Likewise, you need to provide clear information about your outcomes, mortality, and disability. That baseline is in 2012, mortality is from the survey in 2015, thus FU of 3 years (median = 47 months), disability measured by ADL and so on.
All analyses should be stratified by sex because it is an important confounder in the association between SVI and mortality/disability. Please see information about sex differences in ADL in this paper: https://pubmed.ncbi.nlm.nih.gov/32158373/
Please restructure the materials and methods section. 1) Study population 2) Exposure variable, outcome variables, covariates. 3) Statistical methods. Could you provide headlines of these subparts?
Minor comments
Title: Why is it important the highlight in the title that it is a secondary analysis? Please provide a more expressive title. You could mention mortality and ADL and the years 2012 and 2015.
Introduction
- Frailty seems to be very important, but it is “only” a covariate in your analyses. It should be clear from the beginning. It is also given a lot of space in Supp Table 1.
- You need to provide more information about the SVI in relation to lines 44-45. Which social factors? Please elaborate. I suggest you add it here and not (only) in the method section.
- The aim should be clear. Here you must mention that it is mortality with a short follow-up, and you must elaborate what is incident disability.
Materials and methods
- High risk alcohol drinking is strange. Per occasion is always difficult because it is subjective. I am surprised that not more than 17.2% of men and 3.7% of women answers “yes” (you ask for 2 or more drinks per occasion for women and 3 for men). It may be because the amount of drinking in Mexico is low or that people do not understand what per occasion is. There could be different interpretations. This should be added to the discussion section.
- Please explain to the reader what next-of-kin informants are?
Discussion
- You should elaborate the discussion with strengths and limitations. Particularly, is information about mortality reliable? In other surveys such as the Survey of Health, Ageing and Retirement in Europe (SHARE) the mortality data cannot be used because they have not yet validated the mortality data using registers. Can we trust the reliability of the mortality data in this study? How did they find out who died? Is three years of follow-up appropriate? Well-known that you use imputations to take this into account showing similar results this should be elaborated in the paper.
- You should also discuss your results compared with others and discuss what this study adds.
Tables
- Tables 2-4: Should be stratified by sex. If you have too few people in some of the age groups, you could combine them.
- Figure 1: Difficult to distinguish the lines. You can consider dropping this figure otherwise the lines should be easy to distinguish also in black and white.
Reviewer 2 Report
This manuscript tries to analyze the health dynamics of Mexican older adults. It is a secondary analysis of the 2012 and 2015 waves of the Mexican Health and Aging Study. The aim is to describe the association of the SVI with mortality and incident disability in Mexican middle-aged and older adults
I feel it is a well-written, well-structured manuscript, and I outline some minor suggestions for authors to consider
Abstract: please review cut offs symbols
Introduction,
It is well writen, please could the authors add exemples in line 37?
Methods,
Could the authors add a figure describing the lines 72 to 79
Discussion,
Please review line 197: Wallance et al, 2015
Round 2
Reviewer 1 Report
You can accept the paper now. They have answered almost all of my concerns.